# Medication Self-Management in Hospitalised Patients with Schizophrenia or Bipolar Disorder: The Perceptions of Patients and Healthcare Providers

**DOI:** 10.3390/ijerph19084835

**Published:** 2022-04-15

**Authors:** Elke Loots, Josée Leys, Shara Proost, Manuel Morrens, Inge Glazemakers, Tinne Dilles, Bart Van Rompaey

**Affiliations:** 1Department of Nursing Science and Midwifery, Centre For Research and Innovation in Care (CRIC), Nurse and Pharmaceutical Care (NuPhaC), Faculty of Medicine and Health Sciences, University of Antwerp, 2610 Antwerp, Belgium; leys.josee@gmail.com (J.L.); sharaproost@hotmail.com (S.P.); tinne.dilles@uantwerpen.be (T.D.); bart.vanrompaey@uantwerpen.be (B.V.R.); 2Faculty of Medicine and Health Sciences, Collaborative Antwerp Psychiatric Research Institute, University Department of Psychiatry, University of Antwerp, 2610 Antwerp, Belgium; manuel.morrens@uantwerpen.be; 3Faculty of Medicine and Health Sciences, Collaborative Antwerp Psychiatric Research Institute, University Centre for Child and Adolescent Psychiatry Antwerp (ZNA-UKJA), University of Antwerp, 2610 Antwerp, Belgium; inge.glazemakers@uantwerpen.be

**Keywords:** medication self-management, psychiatry, adherence, nursing, prevalence, inpatients, medication errors, qualitative research

## Abstract

Aim(s): The aim of the study was to explore perspectives of hospitalised patients with schizophrenia or a bipolar disorder and their healthcare providers on medication self-management. Methods: In a qualitative descriptive design, semi-structured interviews were used. Forty-nine interviews were completed (nurses *n* = 18; psychiatrists *n* = 3; hospital pharmacists *n* = 2; patients *n* = 26). Data analysis was iterative using an inductive and thematic approach. Results: From the thematic analysis of the interviews, three main themes emerged: monitoring and shared decision-making, relationship based on trust, and patient satisfaction and rehabilitation; as well as three sub-themes: available tools, patient readiness, and safety. Regular monitoring and follow-ups were considered conditions for medication self-management. All stakeholders considered that the patient, the nursing staff, and the psychiatrist should all be involved in the process of medication self-management. All healthcare providers emphasized the importance of regular re-evaluations of the patient and were worried about medication errors and misuse. Most patients considered medication self-management during hospitalisation to increase their confidence, self-reliance, and satisfaction. Many participants thought it would make a positive contribution to the recovery process. Discussion: All stakeholders were positive towards medication self-management under specific conditions. According to the participants, medication self-management offered many benefits, including the implementation of more structure for the patient, an ameliorated preparatory phase towards discharge, and an actual improvement of future adherence. All participants considered medication self-management to contribute to more profound medication knowledge and an overall improvement of their health literacy. Implications and future perspectives: These findings will be used to develop a medication self-management tool in hospitalised patients with schizophrenia or bipolar disorders.

## 1. Introduction

Schizophrenia and bipolar disorders are severe major psychiatric disorders. They are often complicated by recurring relapses [1]. Non-adherence, substance abuse, and stressful life events are risk factors for this relapse, in which non-adherence is the most common cause [2]. Patients interrupting or discontinuing their medication are five times more likely to relapse [3]. Interventions targeting the improvement of adherence in patients with schizophrenia or bipolar disorders are heterogeneous. Current techniques to improve patients’ adherence nearly exclusively use cognitive-behavioral or psycho-education approaches [4,5,6,7,8]. However, about 25% of patients discontinue their medication within the first week after discharge from inpatient treatment [9]. A multidisciplinary approach during hospitalization, focusing on the guidance and coaching of patients in their medication management, seems necessary to prevent relapse. In recent years, the management of chronic illnesses, such as schizophrenia or bipolar disorder, has taken a key place in the patients’ own care process [10,11]. Patients learn to cope more effectively with symptoms, disease, and management of their medication regimens [12,13]. The development and implementation of a medication self-management procedure can facilitate these implementations [14]. Medication self-management has been defined as the extent to which a patient takes medication as prescribed, including not only the correct dose, frequency, and spreading, but also its continued safe use over time [15]. Considering this definition, medication self-management can be deconstructed to identify the patients’ pathway to take medications safely and effectively after hospital discharge [15].

Medication self-management was first mentioned in the literature in 1959, and hence has been internationally studied for many years [16,17,18]. Medication self-management in hospital includes a wide range of activities, such as patient education about medication and monitoring patients while self-managing their medication [19]. In Belgian hospitals, medication self-management is legally allowed under condition of clear registration in the patient’s record and availability of a list of the medications managed by the patient and those managed by the health care provider. The attending physician is responsible for authorizing and evaluating the self-management of the medication process [20]. Medication self-management offers some advantages over administration of medication by nurses, such as increased patient satisfaction and improved adherence to pharmacotherapy and self-care competences [17]. Research conducted in the general hospitals of Flanders’ (the Dutch-speaking region of Belgium) general hospitals showed that 41% of patients (general and surgical units) are capable of medication self-management during hospitalization. Most of these units were medical, surgical, rehabilitation, or geriatric units, not including psychiatric units. A total of 89% of the nurses, 75% of the physicians, and 100% of the hospital pharmacists were willing to allow medication self-management [21]. Little is known about the perception of medication self-management in psychiatry. The aim of this study is to gain insights into the perspectives of all stakeholders involved in the medication self-management procedure in patients with severe mental illness. Insights into the benefits, disadvantages, and conditions of medication self-management during hospitalization are necessary for the development and implementation of a medication self-management intervention. These insights are essential to describe which factors may influence the implementation of a medication self-management procedure.

## 2. Materials and Methods

### 2.1. Research Team and Auhor Reflexivity

EL, TD, and BVR developed the study protocol and topic guide, and EL, JL, and SP conducted the interviews. EL is a doctoral student, researching medication self-management in patients with schizophrenia or bipolar disorders. JL and SP are Master’s students with no affinity with this research. EL, TD, BVR, IG, and MM did the conceptualization, methodology, investigation, and validation. TD, BVR, and MM are the supervisors and have an affinity with this research. IG is a qualitative expert with no affinity with this research project.

### 2.2. Design

In this study, we used a qualitative descriptive design with an exploratory approach within a pragmatic paradigm. The goal was to describe the perceptions around medication self-management during hospitalization from the different stakeholders. Benefits, disadvantages, and conditions for the development and implementation of a medication self-management intervention must be explored. Semi-structured interviews were conducted with hospitalized patients with schizophrenia or bipolar disorders and healthcare providers who were directly involved in patients’ medication process. Findings were synthesized in one comprehensive report on different perspectives. The methods section is described following the consolidated criteria for reporting qualitative studies (COREQ) checklist.

### 2.3. Recruitment

In order to obtain sufficient data variation, convenience sampling was used. We recruited interviewees in four psychiatric inpatient hospitals in Belgium. Units accommodating hospitalized patients with schizophrenia or bipolar disorders were selected. Eligible participants were hospitalized patients with schizophrenia or bipolar disorders and healthcare providers who were directly involved in management of patients’ medication, being a nurse, a psychiatrist, or a hospital pharmacist. Patients were included after consultation with the head nurse and the attending psychiatrist of the unit. Patients had to meet all the following inclusion criteria: adult hospitalized patients diagnosed with schizophrenia or a bipolar disorder type I or II, mentally and physically able to formulate an opinion. Exclusion criteria for patients were patients staying in either an acute or an outpatient unit. In two hospitals, in consultation with the psychiatrist and the head nurse, the researcher informed patients about the study. In the two other hospitals, the psychiatrist and head nurse or team coordinator identified the patients in advance. Subsequently, the first author (EL) personally informed those interested about the study. Healthcare providers were selected after consultation with the director of nursing or head nurse. The healthcare providers were personally invited and informed about the study by the head nurse or researcher considering the structure of each department and the COVID-19 measures. All potential participants were informed personally through an information letter. All eligible participants were informed about the study. Inclusion of new participants was ceased when new information, new ideas, or insights from the interviews no longer emerged and data sufficiency was reached.

### 2.4. Data Collection

The interviews were conducted in Dutch between January 2019 and March 2021. The semi-structured interviews used a topic list based on previous research on medication self-management in hospitalized patients. This process resulted in a topic guide with eight questions (Appendix A). The interview guide was pilot tested with four nurses, three patients, and a psychiatrist, resulting in minor revisions to the content and structure of the guide. The pilot tested interviews were not deleted because nothing was changed in the questions of the topic guide. Only more detailed information on medication self-management during hospitalization at the beginning of the interviews was provided. The pilot interviews showed that participants did not always know well what medication self-management entails. All interviews took place in a quiet room at the unit. Demographical data was noted at the beginning of each interview. All interviews were audio-recorded, and notes were taken during each interview. At the completion of the interview, the interviewer reflected the key-points to the interviewee. No personal or professional relationships existed between the participants and the interviewer prior to the interview. In advance of the interview, participants were informed of the aim and goals of the study.

### 2.5. Data Analysis Process

The data analysis started immediately after the first interview and has been continuously iterative using an inductive and thematic analyzing approach. The data collection and analysis proceeded in parallel. All interviews were transcribed verbatim line by line and cleaned of all identifying information. Microsoft Excel^®^ was used to manage the data. A systematic multistage approach guided this analysis: familiarization, identifying and indexing, mapping, and interpretation [22,23]. After all interviews had been transcribed, data were re-read multiple times to obtain familiarization. Important fragments were assigned to an open, descriptive code which was then converted into an interpretative code. To improve the confirmability of the study, three team members (EL, JL, SP) independently coded four transcripts and compared coding line by line. Any discrepancies were discussed until consensus was achieved. The remaining transcripts were coded independently by the same three team members to identify common high-level concepts. The coding included memos for each transcript and reflections on analysis. The data were discussed at regular intervals to provide consistency in coding. After coding, themes were identified in the mapping stage. We grouped similar codes into those themes and compared themes between patients and healthcare providers. During the interpretation, major themes and associated quotes were identified to summarize the results [22,23,24].

### 2.6. Ethical Considerations

The appropriate local ethics committee and the Ethics Committee of the University Hospital in Antwerp formally granted ethical approval (reference B300202042928). All participants received information on the purpose, design, and execution of the study. Participation was voluntary and signed informed consent was obtained from all participants prior to the interview. Participants had the right to withdraw consent at any time. Participants also agreed with them being audio recorded.

### 2.7. Participants

A total of three psychiatrists, eighteen nurses, two hospital pharmacists, and twenty-six patients were interviewed. In two hospitals, psychiatrists indicated not being able to participate due to lack of time. Interviewing hospital pharmacists was possible in one hospital only. The interviews ranged from 21 to 60 min. On average, an interview with a physician took 28 min (range 23–32 min), with a nurse 45 min (range 27–60 min), with a hospital pharmacist 53 min (range 50–56 min), and with a patient 40 min (range 21–60 min).

Table 1 shows the demographic characteristics of the interviewed participants per hospital.

## 3. Results

### 3.1. Themes

From the thematic analysis of the interviews, three main themes and three sub-themes emerged (Table 2).

#### 3.1.1. Theme 1: Monitoring and Shared Decision-Making

Monitoring and shared decision-making was a frequently discussed item between patients, nurses, and psychiatrists. Regular monitoring and follow-up were considered crucial conditions for medication self-management during hospitalization. Many healthcare providers were concerned about losing an overview or control over the actual medication intake or perhaps not noticing mistakes, overdoses, and/or misuse. Healthcare providers emphasized the daily monitoring to check whether the patient had effectively taken their medication. However, it was noted that there is never absolute certainty about the medication intake, even when administered under supervision. Patients and nurses indicated that patients should always be guided and monitored by healthcare providers during medication self-management. Additionally, the follow-up during and after hospitalization was considered to be important in order to reduce potential medication errors (wrong time or product or improper dose).

Many participants considered the patient, the nurse, and the psychiatrist should all be involved in the process of medication self-management. Specific conditions were described that the organization had to meet in order to organize medication self-management during hospitalization. Participants were convinced that this approach would also enhance medication adherence.


*“Especially in the beginning, I think eh… eh… especially with people who have never done that. Do regular sampling to see if the medication has been taken. Regular monitoring is very important.”*
(Patient 8, bipolar disorder)


*“You have to do some checking every day. You still have to go in the room or everyone’s medication tray in the evening… and look in the bin… are they not in the bin? Have they not been flushed down the toilet?…“*
(Nurse, 16)


*“That supervision and guidance from the nurse is important but you should not last longer than necessary.”*
(Patient 4, schizophrenia)


*“I think that’s also an important part of ‘How do you as a patient see this? Would you like to take it?… ‘Are there any problems? How did it go in the past? How do you feel about taking the medication?’ That this is important, otherwise we will interpret it in the place of the patient.”*
(Nurse 6)

#### 3.1.2. Available Tools

Many participants considered the organization should provide some tools such as the use of pillboxes, medication schedules, electronic reminders, and applications. In addition, psychiatrists and nurses considered that the hospital should provide a medication self-management protocol. Another important condition was how and where to safely store self-management medication. Many participants were concerned about medication abuse or theft of medication. Healthcare providers indicated that providing workshops on medication self-management is an important condition for the implementation of medication self-management. During these workshops, patients could practice medication self-management, ask for tips and tricks, and formulate possible questions. During these exercise sessions, healthcare providers obtained an immediate insight into the patient’s condition.


*“I would perhaps like there to be a kind of community group, where the patients who are almost discharged can go and practice to do their own medication self-management. Uh… a group with stable people with whom we work towards home that we… who have their own living space or something, so that we really are a separate target group actually… in which you can work very intensively with their medication.”*
(Nurse 2)


*“I tell you, with me one of the tools is also setting my alarm clock…there are also those boxes or those things, shall I say, that remind you of that or something…, those little machines…. ”*
(Patient 9, bipolar disorder)

#### 3.1.3. Theme 2: Relationship Based on Trust

All healthcare providers considered it as important to re-evaluate the patients on a regular basis and were worried about medication errors and misuse.

#### 3.1.4. Patient Readiness

Many healthcare providers were concerned about the difficulty of correctly assessing patients’ eligibility for medication self-management. Psychiatrists believed patients are often overestimated, while sometimes being underestimated.


*“I think one of the risks is that people will think ‘He can’t do that’ and so, we don’t do it. That is a risk, that from themselves, there is… It’s very difficult to assess the extent to which people can do things. Sometimes people are chronically overestimated, sometimes they are underestimated.”*
(Psychiatrist)

#### 3.1.5. Safety

The experience of medication side-effects can lead to many discomforts and to stopping the medication with the higher risks as a result, according to patients and nurses. Possible dangers included medication errors, medication intoxication, suicide (attempt), and medication abuse with eventual medical damage to the patient’s health, to others, and to the environment. Despite their willingness to practice medication self-management, patients with schizophrenia were especially anxious and stressed about making medication errors during medication self-management. Patients and nurses were particularly concerned about hoarding the medication.


*“And… yes, maybe from the social aspect, peer pressure, swapping medication, theft… that kind of thing.”*
(Psychiatrist)


*“The threat for the patient is also thinking about it yourself, taking it yourself, yes… and forgetting to take it out of unwillingness, that you forget. It may be that you really forget, it may be that you don’t want to…”*
(Patient 17, bipolar disorder)

#### 3.1.6. Theme 3: Patient Satisfaction and Rehabilitation

Patient satisfaction and rehabilitation were the frequently discussed items.

Nurses, hospital pharmacists, and patients suggested that medication self-management could be beneficial to the hospital and the hospital image due to the potential positive experiences and higher patient satisfaction. Most patients believed medication self-management during hospitalization increased their autonomy, confidence, self-reliance, appreciation, and satisfaction.

Many participants perceived that medication self-management makes a positive contribution to the recovery process. According to the participants, medication self-management during hospitalization offers many advantages, such as more structure for the patients, preparation for discharge and an improvement of adherence.

All stakeholders reported that medication self-management would contribute to better medication knowledge, an improvement of disease insight, and reflection on their own vulnerabilities through the psycho-education and guided training sessions. In addition, nurses suggested that an individual psycho-education program and medication self-management training should be included in the current therapy program for a group of stable patients who are about to be discharged.

Patients considered that they should continue the medication management routines that they used to do at home during hospitalization. This process allowed patients to take their medication at the same time as they were used to at home. Participants were convinced that this approach would also enhance medication adherence.


*“I think that also gives the patient a feeling of, yes… perhaps also of ‘I can do this myself’… ‘I can…’. They also say to me that I can take responsibility for this’, so I think that’s something the patient can be proud of.”*
(Hospital pharmacist 1)


*“By the time, when you come home, you will know how to do it. Yes, then you learn it by the time you get back home.”*
(Patient schizophrenia, 13)


*“The opportunities are that he is more aware of what he is taking, that he has more of a routine, of ‘ah, I have to take my medication’ in the morning and that that would certainly be a good idea in function of going back home after discharge.”*
(Nurse 4)


*“I also think if… he prepares it himself that, he might become more aware of the need for his medication.”*
(Psychiatrist)


*“You would like to do medication self-management. You also see the opportunity of… to be able to do that yourself in preparation to go home, you also say… one week is not enough, it would be better to be able to practice here for three or four weeks in order to be able to go home… to make it your own and build a routine into it… And you believe that this will also benefit the medication adherence.”*
(Patient 17, bipolar disorder)

## 4. Discussion

### 4.1. Main Findings

Most patients already took responsibility for medication prior to admission or shared this responsibility with significant others. In addition, several participants had previous experiences of partial or complete self-management of medication during hospitalization. Overall, it may be stated that medication self-management during hospitalization was found to be very beneficial, especially for patients and for nurses. Results showed differences in patients’ medication self-management views. At the beginning of the interviews, patients with schizophrenia were not eager to medication self-management during hospitalization. When the concept of medication self-management was explained again, with the emphasis on continued support by healthcare providers, they were willing to try medication self-management.

Three main themes were revealed to consider the implementation of a medication self-management tool during hospitalization. Participants reported the importance of monitoring and shared decision-making, a relationship based on trust, and patient satisfaction and rehabilitation. Many of these results aligned with earlier qualitative studies and a systematic review from research in general hospitals [17,25,26]. In studies, the benefits for patients practicing medication self-management during hospitalization resulted in an increased patient confidence before discharge, an improved disease insight, improved medication knowledge, and an improved therapeutic adherence after discharge [17,25,26,27]. Medication self-management during hospitalization would rely on a collaborative and trusting relationship. Moreover, patients might become more confident in medication self-management if they feel supported by nurses. These results aligned with our results. Literature suggested patients were recognized as experts in their own disease management and were able to make decisions about their own care, goals, and values [28]. In our study, psychiatrists noted that medication self-management empowers patients. Medication self-management created possible dangers, such as the possible misuse of medication or increased incidence of medication errors. Currently, a small body of evidence suggests that medication self-management would result in reduced medication errors in a non-psychiatric population [17].

In addition, separately packaged medications in unidoses in the hospitals were seen as an obstacle before and after discharge as they are not user-friendly and not the same as patients’ medication at home. Healthcare providers were particularly concerned about the responsibility of each stakeholder. It was unclear to nurses and patients who has the final responsibility for the health status of patients on admission medication self-management during hospitalization. Therefore, all stakeholders stated that the hospital should have guidelines so that each stakeholder knows their role and responsibility. Many participants reported the importance of the presence of regular monitoring. This monitoring is used not only to prevent abuse, but also to prevent behavioral changes and side effects of medication [29,30].

Previous research reported similar results. Information and communication about pharmacological therapy are important points of attention during the treatment process of patients [31,32,33].

### 4.2. Strengths and Limitations

One of the strengths of this study was the inclusion of the perception of patients, nurses, physicians, and hospital pharmacists situated in four different psychiatric hospitals. The interviews provided valuable in-depth insights into the specific concerns of patients with schizophrenia or bipolar disorder who have complex psychiatric problems requiring medication. The study offers the possibility to develop a medication self-management tool according to the needs of the stakeholders.

There are a few limitations to this study. First, despite our best efforts, we encountered challenges recruiting hospital pharmacists and physicians to participate in this study. We had limited perspectives from hospital pharmacists and psychiatrists, and other professionals may have a different opinion. Second, the participants were selected and addressed by the head nurse or department manager. This method of recruitment could have caused selection bias, as the participating stakeholders were interested in the topic. This way of recruitment is due to the COVID-19 pandemic as access to the units was limited. Finally, according to many patients, significant others and general practitioners are also important stakeholders in the medication process. They should be included in future research.

### 4.3. Implications for Practice

The findings of this study created new opportunities for practice. First, operational strategies can be developed in tools and feasible activities for medication self-management: a patient assessment tool for deciding whether the patient is capable of medication self-management, a monitoring tool for medication intake, and different education sessions and supporting during hospitalization. Second, these strategies may include individual adaptation and simplification of learning and practice opportunities, identification, and management of individual barriers, ensuring patient support structures and improving self-efficacy. Finally, these strategies can form fundamental pillars for the development and testing of a medication self-management toolbox in an intervention study.

## 5. Conclusions

Patients with schizophrenia or bipolar disorder, nurses, psychiatrists, and hospital pharmacists were generally positive about medication self-management during hospitalization, but only under certain conditions. Monitoring and shared decision-making is a much-discussed issue. Patients should be willing to prove themselves during hospitalization and must possess certain competences which are regularly reassessed by healthcare providers. In addition, the organization should offer a procedure and workshops for medication self-management. Many healthcare providers are concerned about the difficulty of correctly assessing patients’ eligibility for medication self-management. Healthcare providers consider it as important to re-evaluate the patients on a regular basis. All participants consider that medication self-management would contribute to better medication knowledge and improvement of their health literacy.

## Figures and Tables

**Table 1 ijerph-19-04835-t001:** Demographic characteristics of the interviewed participants per hospital.

Characteristics (*n* = 49)	Psychiatric Hospital 1	Psychiatric Hospital 2	Psychiatric Hospital 3	Psychiatric Hospital 4	Total Sample
Type of hospital	University psychiatric hospital	Regional psychiatric hospital	Public psychiatric hospital	Private psychiatric hospital	
Number of hospital beds	601	747	263	313	1924
**Codes interviewed groups ***					
Patients					
Schizophrenia	7	1	3	6	17
Bipolar disorder	3	1	3	2	9
Psychiatrists	2	1	0	0	3
Nurses	6	2	4	6	18
Hospital pharmacists	0	2	0	0	2
**Gender (*n*)**					
Patients					
Male	6	1	6	4	17
Female	4	1	2	2	9
Psychiatrists					
Male	2	1	0	0	3
Nurses					
Male	2	1	1	2	6
Female	4	1	3	4	12
Hospital pharmacists					
Female	0	2	0	0	2
**Age patients (years)**	Mean (sd)	Median (min–max)			
	44 (11.6)	43 (26–62)			
**Years working**	Mean (sd)	Median (min–max)			
Psychiatrists	14 (12.5)	13 (2–27)			
Nurses	13 (9.4)	10 (1–27)			
Hospital pharmacists	3.5 (0.7)	3.5 (3–4)			
**Duration of the interviews** **in minutes**	Mean (sd)	Median (min–max)			
Patients	40.3 (12.2)	37 (21–60)			
Psychiatrists	27.5 (4.6)	27 (23–32)			
Nurses	44.6 (9.9)	42.9 (27–60)			
Hospital pharmacists	53 (4.8)	53 (50–56)			

* For each person in each group an individualized code was used.

**Table 2 ijerph-19-04835-t002:** Main and sub-themes.

Theme	Sub-Theme
1. Monitoring and shared decision-making	Available tools
2. Relationship based on trust	Patient readiness
3. Patient satisfaction and rehabilitation	Safety

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
