# Peer review of "Medication Self-Management in Hospitalised Patients with Schizophrenia or Bipolar Disorder: The Perceptions of Patients and Healthcare Providers"

_ijerph, 2022, doi:10.3390/ijerph19084835_

Round 1

Reviewer 1 Report

Thank you for the opportunity to review the revised manuscript Medication self-management in hospitalized patients with schizophrenia or bipolar disorder: the perceptions of patients and healthcare providers.

The manuscript is clearly improved. The number of themes is reduced from five to three and the subthemes are also reduced, however, the latest could be reduced even more.

In the result section, it could be clearer “who says/thinks what”. When the term “participants” is used, it is difficult to know who the authors are referring to.

I suggest a further revision of the result and discussion section.

Abstract:

Results: Usually the main themes are presented without “( )”. I suggest deleting the “( )”.

Line 32 – 37: How are these findings related to the themes? It would be beneficial if this was clearer. I suggest that the content of each main theme is described briefly.

Line 37-41: Is “discussion” meant as the conclusion of the study? If so, I suggest it be stated more clearly. Line 37-38 can be left out, as it does not provide any information in relation to the content of the study.

Manuscript

Line 167-168: “All interviews were audio-recorded” is a repetition from the section above and can be left out.

Line 191: Delete “setting”. I suggest that “population” is replaced with “participants”. As mentioned in my first review, I suggest that population/participants are presented in the method section, which is the most common in qualitative study designs. In the abstract, participants are presented in the method section.

Line 213 - 217: It could be nice with a brief introduction to the content or dilemma of the theme. The same applies to theme 2.

There is no need for headlines or naming subthemes for “monitoring and follow-up by healthcare providers” and “Shared decision-making”, as this is contained in the main themes. The presentation of the results becomes very abrupt when it is presented like this.

Line 227: “Participants” Who are they?

Line 256 – 269: This is too short a section to be a subtheme, as it is very superficial described. See note above.

Line 270: It is not clear how the name of the main theme embraces this subtheme. It is difficult to see the connection. Could it somehow be merged with the “monitoring” issue?

Line 405: “outcome” is often related to quantitative research designs. The name of the theme does not indicate the content of the theme, which is preferable. A suggestion could be to name the theme “patient satisfaction and rehabilitation” and then delete/not naming any subthemes.

Discussion

I suggest a more structured and specific/detailed discussion. In the first paragraph of the discussion section, I need a short summary of your findings and a presentation of what you will discuss in the discussion section.

I suggest that the discussion section is reversed.

Line 490 – 492: Suggest to be deleted.

Line 492 – 493: Move to “implications for practice”

Line 493 – 496: Suggestion: Delete or move to “strength or limitation”.

Line 512 - 513: Suggest to be deleted.

Line 516 – 517 (+ until 532): This is a far too broad a statement. You must go into details about which of your findings are exactly the same/or the opposite as findings from other studies. You cannot discuss all your themes and findings in the same section.

After revision of the manuscript, the conclusion might need some adjustment.

Author Response

Comments from reviewer #1:

Thank you for the opportunity to review the revised manuscript Medication self-management in hospitalized patients with schizophrenia or bipolar disorder: the perceptions of patients and healthcare providers.

The manuscript is clearly improved. The number of themes is reduced from five to three and the subthemes are also reduced, however, the latest could be reduced even more.

In the result section, it could be clearer “who says/thinks what”. When the term “participants” is used, it is difficult to know who the authors are referring to.

I suggest a further revision of the result and discussion section.

Abstract:

(Query): Results: Usually the main themes are presented without “( )”. I suggest deleting the “( )”.

Line 32 – 37: How are these findings related to the themes? It would be beneficial if this was clearer. I suggest that the content of each main theme is described briefly.

Line 37-41: Is “discussion” meant as the conclusion of the study? If so, I suggest it be stated more clearly. Line 37-38 can be left out, as it does not provide any information in relation to the content of the study.

 (Response): Thank you for the suggestions. We understand your remarks and we have revised the abstract.

“Results: From the thematic analysis of the interviews, 3 main themes emerged: Monitoring and shared-decision making, Relationship based on trust and Patient satisfaction and rehabilitation; as well as 3 sub-themes: available tools, patient readiness and safety. Regular monitoring and follow-up were considered conditions for medication self-management. All stakeholders considered that the patient, the nursing staff and the psychiatrist should all be involved in the process of medication self-management. All healthcare providers emphasized the importance of regular re-evaluations of the patient and were worried about medication errors and misuse. Most patients considered medication self-management during hospitalisation to increase their confidence, self-reliance and satisfaction. Many participants thought it to make a positive contribution to the recovery process. Discussion: All stakeholders were positive towards medication self-management under specific conditions. According to the participants, medication self-management offered many benefits, including the implementation of more structure for the patient, an ameliorated preparatory phase towards discharge and an actual improvement of future adherence. All participants considered medication self-management to contribute to more profound medication knowledge and an overall improvement of their health literacy. Implications and future perspectives: These findings will be used to develop a medication self-management tool in hospitalised patients with schizophrenia or bipolar disorders.”

Manuscript

(Query): Line 167-168: “All interviews were audio-recorded” is a repetition from the section above and can be left out. Line 191: Delete “setting”. I suggest that “population” is replaced with “participants”. As mentioned in my first review, I suggest that population/participants are presented in the method section, which is the most common in qualitative study designs. In the abstract, participants are presented in the method section.

(Response):
 Thank you for this suggestion. We took notice of your remarks and we have amended the manuscript accordingly.

(Query): Line 213 - 217: It could be nice with a brief introduction to the content or dilemma of the theme. The same applies to theme 2.

 (Response): We have modified the entire results section and we wrote a short introduction on the content of theme 1 and 2.

There is no need for headlines or naming subthemes for “monitoring and follow-up by healthcare providers” and “Shared decision-making”, as this is contained in the main themes. The presentation of the results becomes very abrupt when it is presented like this.

 (Response): Thank you for this suggestion. We have merged the sub-themes “monitoring and follow-up by healthcare providers” and “Shared decision-making” into the main theme. This should be more clear now.

Theme 1: Monitoring and shared-decision making

Monitoring and shared-decision making was a frequently discussed item between patients, nurses and psychiatrists. Regular monitoring and follow-up were considered crucial conditions for medication self-management during hospitalisation. Many healthcare providers were concerned about losing an overview or control over the actual medication intake or perhaps not noticing mistakes, overdoses and/or misuse. Healthcare providers emphasised the daily monitoring to check whether the patient had effectively taken his medication. However, it was noted that there is never absolute certainty about the medication intake, even when administered under supervision. Patients and nurses indicated  that patients should always be guided and monitored by healthcare providers during medication self-management. Also, the follow-up during and after hospitalisation was considered to be important in order to reduce potential medication errors.

Many participants considered the patient, the nurse and the psychiatrist should all be involved in the process of medication self-management. Specific conditions were described that the organisation had to meet in order to organise medication self-management during hospitalisation. Participants were convinced that this approach would also enhance medication adherence.

(Query): Line 227: “Participants” Who are they?

 (Response): Indeed, thank you for this feedback. We have amended the other sentences in the manuscript:

Patients and nurses reported patients should always be guided and monitored by healthcare providers during medication self-management. Also, the follow-up during and after hospitalisation was important possibly reducing medication errors.

(Query): Line 256 – 269: This is too short a section to be a subtheme, as it is very superficial described. See note above.

(Query): Line 270: It is not clear how the name of the main theme embraces this subtheme. It is difficult to see the connection. Could it somehow be merged with the “monitoring” issue?

(Response): We have modified the entire results section and we merged the sub-theme “adherence” with Theme 3: Patient satisfaction and Rehabilitation.

Theme 3: Patient satisfaction and Rehabilitation

Patient satisfaction and rehabilitation were the frequently discussed items.

Nurses, hospital pharmacists and patients suggested that medication self-management could be beneficial to the hospital and the hospital image due to the potential positive experiences and higher patient satisfaction. Most patients believed medication self-management during hospitalisation increased their autonomy, confidence, self-reliance, appreciation and satisfaction.

Many participants perceived that medication self-management makes a positive contribution to the recovery process. According to the participants, medication self-management during hospitalisation offers many advantages, such as more structure for the patients, preparation for discharge and an improvement of adherence.

All stakeholders reported that medication self-management would contribute to better medication knowledge, an improvement of disease insight and reflection on their own vulnerabilities through the psycho-education and guided training sessions. In addition, nurses suggested an individual psycho-education programme and medication self-management training, should be included in the current therapy programme for a group of stable patients who are about to be discharged.

Patients considered that they should continue the medication management routines that they used to do at home during hospitalisation. This process allowed patients to take their medication at the same time as they were used to at home. Participants were convinced that this approach would also enhance medication adherence.

(Query): Line 405: “outcome” is often related to quantitative research designs. The name of the theme does not indicate the content of the theme, which is preferable. A suggestion could be to name the theme “patient satisfaction and rehabilitation” and then delete/not naming any subthemes.

(Response): Thank you for this suggestion. We have modified the entire results section and we reworked and renamed theme 3 to patient satisfaction and rehabilitation.

Discussion

I suggest a more structured and specific/detailed discussion. In the first paragraph of the discussion section, I need a short summary of your findings and a presentation of what you will discuss in the discussion section.

(Query): I suggest that the discussion section is reversed.

Line 490 – 492: Suggest to be deleted.

Line 492 – 493: Move to “implications for practice”

Line 493 – 496: Suggestion: Delete or move to “strength or limitation”.

Line 512 - 513: Suggest to be deleted.

Line 516 – 517 (+ until 532): This is a far too broad a statement. You must go into details about which of your findings are exactly the same/or the opposite as findings from other studies. You cannot discuss all your themes and findings in the same section.

(Response):  Thank you for these suggestions. We have amended all the above sentences in the manuscript.

Reviewer 2 Report

I think that minor stylistic improvements can still be made. Here is a list of suggestions:

Medication self-management was first mentioned in THE literature in 1959 (72)

reseach conducted in the general hospitals of Flanders – the Dutch-speaking region of Belgium – etc/ (83)

in patients with severe mental illness … > when patients suffer from severe mental illness (92)

In only one hospital, it was possible to interview hospital pharmacists.> Interviewing hospital pharmacists was possible in one hospital only (198)

Patients considered it was important that they could continue they could continue medication management routines they used to do at home during hospitalisation > that they should continue the medication management routines that they used to do at home while being hospitalized (309)

while sometimes underestimated > whil sometimes being underestimated (339)

one of the threats is that… > one of the risks / dangers (342). Same line (362).

with the higher risks as a result (350)

were the two most discussed items > the two most widely / frequently discussed topics / themes (408).

This study explores the different points of view expressed by patients, psychiatrists and pharmacists regarding medication self-management in patients with schizophrenia or bipolar disorder. The findings may eventually contribute to the development and implementation of a medication self-management tool in psychiatric hospitals. (490-493)

In order to implement medication self-management in psychiatric hospitals, it is necessary to explore the different points of view of involved stakeholders > SHOULD BE DELETED (579)

Patients with schizophrenia or bipolar disorder, nurses, psychiatrists and hospital pharmacists being generally positive about medication self-management during hospital- isation provided several conditions > Patients with schizophrenia or bipolar disorder, nurses, psychiatrists and hospital pharmacists are generally positive about medication self-management during hospital- isation but only under certain conditions (581-83)

Balancing between monitoring and shared-decision making was a much discussed item. > Balancing between monitoring and shared-decision making is a much discussed issue. (583)

Lines (587-91) : USE THE PRESENT TENSE (SINCE YOUR AIM IS TO RECAPITULATE THE OPINIONS HELD BY THE PARTICIPANTS). ALSO, YOU ARE IMPLICTLY TRYING TO EXTEND YOUR FINDINGS TO SIMILAR HEALTHCARE ENVIRONMENTS.

Author Response

Comments from reviewer #2:

(Query): I think that minor stylistic improvements can still be made. Here is a list of suggestions:

-Medication self-management was first mentioned in THE literature in 1959 (72)

-research conducted in the general hospitals of Flanders – the Dutch-speaking region of Belgium – etc/ (83)
in patients with severe mental illness … > when patients suffer from severe mental illness (92)
-In only one hospital, it was possible to interview hospital pharmacists.> Interviewing hospital pharmacists was possible in one hospital only (198)

-Patients considered it was important that they could continue they could continue medication management routines they used to do at home during hospitalisation > that they should continue the medication management routines that they used to do at home while being hospitalized (309)

-while sometimes underestimated > while sometimes being underestimated (339)

-one of the threats is that… > one of the risks / dangers (342). Same line (362).

-were the two most discussed items > the two most widely / frequently discussed topics / themes (408).

(Response):  Thank you for these suggestions. We have amended all the above sentences in the manuscript.

(Query): This study explores the different points of view expressed by patients, psychiatrists and pharmacists regarding medication self-management in patients with schizophrenia or bipolar disorder. The findings may eventually contribute to the development and implementation of a medication self-management tool in psychiatric hospitals. (490-493)

 (Response): We understand your remark and we have revised the sentence.

(Query): In order to implement medication self-management in psychiatric hospitals, it is necessary to explore the different points of view of involved stakeholders > SHOULD BE DELETED (579)

 (Response): Indeed, thank you for this feedback. We have amended the other sentences in the manuscript.

(Query): Patients with schizophrenia or bipolar disorder, nurses, psychiatrists and hospital pharmacists being generally positive about medication self-management during hospital- isation provided several conditions > Patients with schizophrenia or bipolar disorder, nurses, psychiatrists and hospital pharmacists are generally positive about medication self-management during hospital- isation but only under certain conditions (581-83)

Balancing between monitoring and shared-decision making was a much discussed item. > Balancing between monitoring and shared-decision making is a much discussed issue. (583)

Lines (587-91) : USE THE PRESENT TENSE (SINCE YOUR AIM IS TO RECAPITULATE THE OPINIONS HELD BY THE PARTICIPANTS). ALSO, YOU ARE IMPLICTLY TRYING TO EXTEND YOUR FINDINGS TO SIMILAR HEALTHCARE ENVIRONMENTS.

 (Response): Thank you for these suggestions. We understand your remarks and we have revised the sentences.

This manuscript is a resubmission of an earlier submission. The following is a list of the peer review reports and author responses from that submission.

Round 1

Reviewer 1 Report

I am afraid I am not the best judge of your work. I was trained as a cognitive linguist and conversation analyst. When I was offered to review your paper, I thought more attention would be paid to the wording and  discourse strategies used by the interviewees. The main concern, it would seem, was not expression but content: identifying the 5 main "themes" and 10 "subthemes" that underlie the discourse produced by the nurses, doctors, pharmacists and patients that you interviewed.

I do understand of course that priority was given to construal and "perception" not conversational style. I am also  aware that working on English translations (adaptations?) of talk originally produced in Dutch (Flemish) poses numerous challenges. So I am in no way suggesting that your paper is "flawed" but that my "expertise" is of very little relevance.  I wish I could provide greater insight and help you polish things up.

Something that definitely needs to be improved is grammatical correctness. Many sentences are stylistically awkward (overly complex) or gammatically incorrect. I would also suggest revising the absract. You should  ask one or two native speakers of English - preferably in your field - willing to assist you with those necessary corrections. Please avoid repeating "medication self-management during hospitalization in patients with schizophrenai or bipolar disorder" over and again in your piece.  (E.g. The sentence is used twice in the abstract).

Your enquiry is more than just a survey. The precious material you collected should indeed provide precious material to design the "self-management tools" mentioned at the end of the study, although too little precision is given.

Slips and mistakes (sampled)

E.g. (18) data analyse (analysis?) - Check the meaning of "threats" in English. I don't think the word carries what you actually mean - (125) did not always knew well (know) - (205) patients should always be guiding (guided?) - (225) that supervision and guidance from the nurse is necessary but you should not see it longer than necessary (it should not last longer than necessary) - (245) the organization should be obliged to provide (should provide) - (278) at long term (in the long run) - (287)  some threets for medication self-management [what do you mean by "threats"?] - (347) by the time when you come home (by the time you come home / get back home) - (359)  the more  that you can make the patient responsible (the more you make the patient responsible / act responsibly) - (398) this study explored insights into the perspective of patients  [very awkward] - (481) * to explore into different points of view [delete 'into'

Reviewer 2 Report

Thank you for the opportunity to review the paper Medication self-management in hospitalized patients with schizophrenia or bipolar disorder: the perceptions of patients’ and healthcare providers’. The topic of medication self-management in hospitals is highly relevant, and there is a lack of knowledge about medication self-management in psychiatric settings.

The study’s strength is the inclusion of patients and a variety of healthcare professionals from four different psychiatric settings. In addition, the number of included participants is very satisfying for a qualitative study design.

A major limitation of the study is the superficial presentation of the results of the analysis of data. The study states to use an inductive and thematic approach and presents five themes and ten subthemes, which is a lot for a qualitative study design. The many themes and subthemes are part of the explanation for the superficial presented results. The authors should have made choices according to the most important themes/results and presented them more coherently. Alternatively, some of the themes/subthemes could be merged to create a more coherent presentation of the results.

The result section also presents too many quotes. A quote must show a point of the study findings. More than one quote about the same issue is redundant.

It can be questioned whether the study uses an inductive approach or if it is a deductive approach – or is it a mixed approach? For example, were the themes treats, and benefits part of the interview guide and afterward turned into a theme?

Below are more specific comments to the manuscript.

ABSTRACT

The number of interviews and participant characteristics are presented in the method section and not in the result section as expected in a qualitative research paper.

Data analysis is described as an inductive and thematic approach; therefore, it is expected that the result section presents which themes have been uncovered. This could be more clear.

INTRODUCTION

Line 72-73        The sentences appear a little isolated. Perhaps they could be written in greater context with the text above?

MATERIALS AND METHODS

Line 80-82        “Patients should be able…..” Who says that? If it is the authors’ opinion? I think it should be left out here. There can be argued for this statement in the background section or the discussion section.

Line 96              “Sufficiency…” what does that mean here?

Line 119            I have no access to appendix S1 and thereby the interview guide, which is essential for assessing the study, as the topics and approach during the interview are not described elsewhere.

RESULTS

Line 159ff         Setting and population are usually presented in the method section in qualitative studies.

Line 167-170   The same information is provided in Table 1 and be left out in one of the places.

Line 176            Five themes and ten subthemes emerged from the analysis. This is a lot in a qualitative context. A risk is that the themes are handled superficially with so many themes.

Line 189-196    The two quotes represent the same point – the patient has to be stable/able; one quote is enough.

Line 209-229    A lot of quotes are presented. How do the authors interpret these quotes? What do the authors want the readers to understand/get insight into? Do they represent the same point or different points? A reduction and explanation of quotes are needed.

Line 272-285    “Time management” – a very simple managed theme.

Line 290-298    “Patient readiness assessment”. Again, a short (sub-)theme. Is the theme about the theme of patients being ready for self-management?

Line 309-319    Three quotes - on the same matter? Do the quotes express three different opinions? What do the authors what the reader to make of these three quotes?

Line 329-339    Same comment as earlier about more than one quote after the other.

Line 344-364    Same comment as earlier about more than one quote after the other.

Line 383-396    Same comment as earlier about more than one quote after the other.

DISCUSSION

Line 406            “concluded” might not be the optimal term to use here in the discussion section.

Line 409-410    “Patients with schizophrenia…. were not analyzed in this study”, I think a rephrase is needed.

Line 413-417    Is this more a finding/result of the study?

Line 423-425    “Many of the benefits, threats, opportunities, and conditions…. aligned with earlier … studies”. This is a comprehensive statement. Which findings exactly are the same as other findings?

Line 425-428    What is the connection between the statement and the references? This must be clarified.     

Line 457            Why is it a strength that the study took place during the COVID-19 pandemic?

CONCLUSION

The conclusion might need to be adjusted a bit after a revision of the manuscript.

Reviewer 3 Report

Loots et al. report about perspectives of hospitalised patients with schizophrenia or a bipolar disorder and their healthcare providers on medication self-management during hospitalisation. From total 49 interviews, authors mentioned medication self-management during hospitalisation offers many benefits such as more structure for the patients, preparation for discharge and an improvement of adherence.

Here is my comments for authors considerations;

  1. This is survey-interview based study. Rational for study is not clear. In Hospitals already guidelines are available with expert psychiatrist. Only opinion by interview will not make any change.
  2. Population is much too small to draw any conclusions. Total 49 interviews out of with only 26 patients.  The authors should have compared to population databases instead.
  3. Line 72, "Physicians, pharmacists and nurses were positive towards medication self-management (21)". This is outcome of study. Why you mentioned reference here?
  4. Proper statistical analysis is missing. 
